# Navigating Research Challenges: Collaborative Insights from a Research Retreat During a Healthcare Emergency in Puerto Rico

**DOI:** 10.3390/ijerph22040623

**Published:** 2025-04-16

**Authors:** Katherine Matos-Jiménez, Natalie Alamo-Rodriguez, Emma Fernández-Repollet

**Affiliations:** 1Research Centers in Minority Institutions (RCMI) Program, Center for Collaborative Research in Health Disparities (RCMI-CCRHD), University of Puerto Rico, Medical Sciences Campus (UPR-MSC), San Juan, PR 00936, USA; katherine.matos1@upr.edu; 2Comprehensive Cancer Center, University of Puerto Rico, San Juan, PR 00936, USA; natalie.alamo1@upr.edu

**Keywords:** COVID-19, health disparities, research challenges, remote work, healthcare emergency, professional development

## Abstract

Puerto Rico has faced significant damage from natural disasters and the COVID-19 pandemic, disrupting clinical services and scientific research logistics. In response, the RCMI-CCRHD Program at the UPR-MSC organized a retreat with the objective of understanding the challenges faced by its research network during the pandemic and strategies to overcome them. The retreat featured presentations on COVID-19 supplemental projects and included a discussion group. Twenty attendees identified the challenges they encountered and the strategies developed through an open-ended question and a satisfaction survey, implementing a mixed-method approach. We performed a content analysis on the open-ended survey questions and used descriptive statistics for the satisfaction survey. Key challenges included remote work implementation, recruitment difficulties, and mental health concerns. Attendees shared actions taken to address these issues, such as modifying protocols for remote planning and using digital platforms for improving communication. They also recognized opportunities that arose from remote work, which allowed them to focus on publishing and adapting mental health support tools. The retreat received positive feedback, with 93.8% of attendees giving a five-star rating. By reflecting on these experiences, investigators can refine strategies and implement effective solutions. Recommendations include flexible IRB protocols, improved digital tools, community involvement, and robust emergency preparedness plans.

## 1. Introduction

### 1.1. COVID-19 in Puerto Rico

Puerto Rico has recently faced numerous environmental and societal challenges, including Hurricanes Irma and Maria (2017), ongoing seismic activity (2020), and the COVID-19 pandemic, which began in March 2020. These events have significantly strained the island’s social and economic systems, healthcare infrastructure, and scientific research logistics. The COVID-19 pandemic, in particular, exacerbated existing vulnerabilities, prompting the Government of Puerto Rico to issue Executive Order No. 2020-20 [1,2,3]. This declaration of a state of emergency implemented a lockdown that included a 10-h curfew, restrictions on gatherings and daily activities, travel limitations, and the closure of nonessential businesses [2,4]. It also mandated the abrupt closure of all public and private schools and universities, forcing a transition to online learning formats. To mitigate the effect of this pandemic, the Government of Puerto Rican created a Medical Task Force that included a group of scientists and health professionals to manage data regarding COVID-19 and facilitate decision-making [5]. This task force was also responsible for educating the community about symptoms, treatments, and preventive techniques [5].

The administration of the COVID-19 vaccine in Puerto Rico began on December 15, 2020, prioritizing healthcare workers and populations at greater risk of severe disease [6]. The University of Puerto Rico, Medical Sciences Campus (UPR-MSC), became one of the first vaccination centers on the island [6]. In July 2021, the government implemented strict vaccination mandates, contributing to Puerto Rico’s status as one of the U.S. territories with the highest vaccination rates [6,7]. Despite these efforts, as of 31 July 2024, the Puerto Rico Department of Health reported 517,884 confirmed COVID-19 cases and 7635 related deaths, with only 3.4% of the population fully vaccinated [7].

The UPR-MSC played a pivotal role in addressing the pandemic, spearheading initiatives to collect data, administer COVID-19 tests, and distribute vaccines. Researchers collaborated to develop the infrastructure necessary to implement these programs, aligning these efforts with the institution’s strategic plan to sustain campus operations and advance research on COVID-19 treatments and vaccines. For years 2020–2021, the National Institute of Health (NIH) RePORTER documents 16 projects, nine sub-projects, 428 publications, and two clinical studies on COVID-19 led by researchers at the University of Puerto Rico Medical Sciences Campus [8].

### 1.2. Impact of COVID-19 on Institutional Facilities

The COVID-19 pandemic significantly disrupted educational institutions, leading to the suspension or cancellation of teaching and research activities [9]. The government of Puerto Rico issued over 100 executive orders aimed at controlling the virus, which included directives for the necessary closure of both governmental and private sector operations [10] to encourage isolation as a positive health behavior. Many institutions had to reduce maintenance services and limit on-campus operations to comply with health guidelines, with long-term consequences for their facilities. Health concerns became a primary focus as institutions rapidly transitioned to online tools [11] and implemented new safety protocols for those on campus. This transition presented challenges, as many institutions struggled to provide the necessary training and resources for faculty and students.

### 1.3. Impact of COVID-19 on Research Activities

The pandemic disrupted research timelines, particularly for projects requiring fieldwork, and hindered collaborations due to restricted access to essential facilities [12,13]. Academic leaders, especially department chairs, navigated heightened complexities in balancing administrative responsibilities, research demands, and crisis management [14]. Higher education institutions developed effective strategies to sustain operations, with the transition to online learning emerging as a significant milestone [9,15,16]. This shift fostered several benefits, including enhanced technical skills among staff, new online learning techniques, improved feedback mechanisms, and progress in digital research activities [17,18,19].

In Puerto Rico, academic institutions demonstrated resilience by creating a “new normal” to meet internal and external demands with limited resources. The compounding effects of health-related challenges placed Puerto Rican academic institutions at the center of adaptation and restructuring their policies and practices [19]. The efforts leading to achieving these policies and practices are detailed in the publication *Institutional Resilience in Puerto Rico: A first look at efforts from the Puerto Rican HSIs* [19].

The Research Centers in Minority Institutions (RCMI) Center for Collaborative Research in Health Disparities (CCRHD), was actively involved in the research activities conducted at the University of Puerto Rico Medical Sciences (UPR-MSC) during the COVID-19 pandemic. The focus of the RCMI-CCRHD on health disparities allowed for studies and interventions relevant to underserved communities at a critical time. In an effort to learn from those experiences, a special retreat was organized to identify the challenges faced by the RCMI-CCRHD investigators and staff while conducting health disparities research during the pandemic and the strategies developed to address such challenges. The research question we aim to address through this activity is: What challenges did researchers at the UPR-MSC—affiliated with the RCMI-CCRHD program—encounter during the COVID-19 pandemic, and what strategies were employed to overcome these challenges? This report summarizes the results of this endeavor and underscores the importance of adaptation in sustaining research amid unprecedented disruptions.

This study offers unique insights by focusing on the dual challenges faced by academic researchers in Puerto Rico, a region that has experienced compounding disruptions due to natural disasters and the COVID-19 pandemic. While the existing literature highlights the global impacts of the pandemic on academic research, this work provides a localized perspective, particularly within a minority-serving institution. The incorporation of a retreat-based approach allowed for a collaborative exchange among researchers, fostering the identification of practical, context-specific strategies to address these challenges. The study examines the negative impacts and highlights opportunities that arose from these difficulties, offering useful lessons to improve resilience and preparedness in the future.

## 2. Materials and Methods

### 2.1. Retreat Activity

In December 2023, the RCMI-CCRHD Program administration core organized the retreat “Navigating Unforeseen Challenges: Lessons for Tomorrow” with the aim of discussing the challenges faced while conducting research activities during the COVID-19 pandemic and the strategies adopted to address them. The retreat featured oral presentations from two COVID-19 supplemental research projects: the “COVID-19 Vaccine Uptake Study” and the “Impact of COVID-19 on Maternal Health in Puerto Rico”. These projects were led by early-stage investigators. An additional presentation, “Communicating Science to Lay Audiences”, focused on effective communication strategies emerging from experiences during the COVID-19 pandemic. All oral presentations addressed the research challenges experienced during the COVID-19 pandemic and the actions taken to overcome them.

### 2.2. Design

We employed a mixed-methods approach. Following the presentations, a group discussion session was held, during which attendees were divided into six groups. In this session, attendees completed an open-ended survey identifying the main challenges they faced during the pandemic and the strategies that they developed to address them. At the end of the activity, attendees filled out a survey to assess their overall satisfaction with the retreat.

### 2.3. Attendees

The RCMI-CCRHD research network, including investigators, administrators, coordinators, and technical staff, was invited to participate in the retreat. We employed a convenience sampling method, with eligibility criteria requiring participants to be affiliated with the RCMI-CCRHD program and actively involved in research activities during the COVID-19 pandemic (2020–2022). Invitations were sent via a promotional flyer distributed through the Program’s institutional email list. A total of 20 research staff affiliated with the RCMI-CCRHD accepted the invitation to participate in the activity. For the group session, attendees were randomly seated at the available tables.

### 2.4. Survey Setting

The University of Puerto Rico, Medical Science Campus’s Institutional Review Board exempted this activity from the informed consent process due to the absence of personal identification data collection. Twenty attendees participated in the retreat and provided their responses. They were randomly divided into six discussion groups and completed two anonymous surveys. Data were collected through these surveys, with no distinctions made regarding the affiliation of attendance positions under the RCMI-CCRHD or discussion group. The first survey explored the challenges encountered in research activities during the pandemic, while the second assessed overall satisfaction with the activity. The presenters from the oral presentations, as outlined in Section 2.1, also provided their responses to be included in our measurements.

The first survey, which is included in the Appendix A, comprises three open-ended questions aimed at exploring research activities related to the following themes: (1) Challenges, (2) Actions Taken, and (3) Emerging Opportunities. This survey was developed by the principal investigator and director of the RCMI program. It did not go through a validation process. Attendees identified the main challenges faced during and after the pandemic, the actions taken to minimize these challenges, and new opportunities that emerged from the experience on an answer sheet. The attendees also evaluated the retreat on Survey Monkey’s online platform, rating their satisfaction with the overall activity, invited presentations, and group discussion. The satisfaction questionnaire, which is included in the Appendix A, was developed by the program coordinator and evaluation leader. It was constructed considering the standardized questionnaires used to evaluate all activities and seminars offered as part of the program. The questionnaire assessed various aspects, including satisfaction with coordination, allocation, agenda, professional value related to the activity, presentation quality, and overall rating of the activity. Three five-point Likert scales: “Completely Agree, Agree, Disagree, Completely Disagree, and Not Applicable”, “Excellent, Good, Fair, Poor, and Not Applicable”, and a 5-star rating scale were used to assess their ratings.

### 2.5. Analyses

The attendees’ responses were analyzed using qualitative content analysis via the Dedoose platform (software version number 9.0.54), which facilitated the identification of themes among the responses, including frequency and total repetition percentage. To conduct this analysis, we first developed a codebook (Appendix A) that included the categories with their corresponding definitions. We used both inductive and deductive processes to generate these categories. Preliminary categories were generated according to the question guide, but after reading the responses additional categories emerged. The codebook, included in the Appendix A, comprises a total of 26 categories organized into three main sections. The first section, the Challenges Category, includes categories such as remote work, incentive distribution, protocol revisions, and mental health. The second section, Actions Taken Category, outlines categories such as modifying protocols, knowing your community’s needs, and the use of digital platforms and technologies. Finally, the Emerged Opportunities Category highlights new development categories, such as new protocols, enhanced knowledge of technological tools, participants’ new reach, and new incentive distribution systems.

Next, we proceeded to code the responses. Two members of the research team independently coded the responses, followed by a coder agreement process. When discrepancies arose between the codes, they were discussed until a consensus was reached. These codes were then uploaded to the Dedoose program to determine the frequency and percentage of category repetition. Satisfaction Survey responses were assessed by calculating each response rating and percentage of the corresponding scales.

## 3. Results

### 3.1. Group Activity Responses

The group discussion session achieved a 100.0% response rate, with 20 attendees completing an open-ended survey identifying their main challenges during the pandemic. The distribution of affiliation of the attendees’ positions under the RCMI-CCRHD included eight technicians (40%), six investigators (40%), three pilot project investigators (15%), two administrative staff members (10%), and one student (5%), no distinctions on the responses were made regarding this status. A more comprehensive response was achieved by following up after the retreat meeting. The COVID-19 pandemic presented significant unforeseen challenges for research teams, with remote work affecting 65.0% of attendees and diminished social interactions affecting mental health (45.0%). An additional perceived obstacle was the recruitment limitations experienced when conducting research studies (45.0%).

In response to these challenges, researchers and staff reported implementing a series of strategies and actions to address them. A significant majority (70.0%) utilized digital platforms and technologies and modified study protocols (60.0%), introducing a secure distribution mechanism of incentives to connect with research staff and study subjects. Knowing the community needs (30.0%) was also a reported action taken to adapt to the pandemic effects on the research studies.

Despite these challenges, the pandemic presented several opportunities to address similar unforeseen events better. The lessons learned included the better use of social media to maintain contact with staff and study subjects (70.0%), more time to write articles (45.0%) and identify funding opportunities (40.0%), and developing the tools needed for protecting the mental health of support personnel (15.0%). Notably, attendees enhanced their knowledge of technology tools and social media platforms, emphasizing a shift towards more technology-driven approaches in research activities (See Table 1). Examples of qualitative results from attendees’ responses are shown in Table 2 to offer further insight into their experiences and perspectives.

### 3.2. Research Retreat Evaluation

The research retreat survey received a 100.0% response rate, with 20 attendees providing feedback on their overall satisfaction with the activity. The retreat featured presentations and activities that earned positive feedback from the attendees. As shown in Table 3, the oral presentations from two COVID-19 supplemental research projects received high ratings: “COVID-19 Vaccine Uptake Study” was rated as excellent by 80.0% of attendees and “Impact of COVID-19 on Maternal Health in Puerto Rico” received an 80.0% excellent rating. The third presentation, “Communicating Science to Lay Audiences”, was rated as excellent by 85.0% of attendees (refer to Table 3).

In addition, the retreat contributed to professional development, with 75.0% of attendees expressing complete agreement that it had a positive impact. The organization and logistics of the event were also well-received, with unanimous agreement regarding the effectiveness of the coordination as well as the adequacy of the agenda, location, and time allocation (see Table 4). The overall retreat rating was 93.8% five-star rating.

## 4. Discussion

The attendees’ challenges were primarily related to the control measures enacted in response to the COVID-19 pandemic. These measures included a series of government-mandated lockdowns implemented by institutions in Puerto Rico to mitigate the spread of the virus. As stated by UNESCO, the pandemic disrupted research timelines, particularly for projects requiring fieldwork, and hindered collaborations due to restricted access to essential facilities [11]. This aligns with the findings of our retreat, highlighting how such restrictions can significantly impact research activities and study subjects’ engagement. Among the various challenges reported, remote work emerged as the most significant issue, greatly affecting adherence to the previously established research protocol. This is consistent with existing literature emphasizing the difficulties of maintaining research continuity in a remote environment. Additionally, the isolation and limited mobility experienced by many during this time created substantial barriers to recruitment and retention efforts, further corroborating the challenges outlined by UNESCO regarding the overall impact of the pandemic on research operations. Recognizing these constraints is essential for understanding the broader implications of the pandemic on the research landscape.

In discussing the challenges encountered during the survey, it is essential to emphasize the significant mental health burden many attendees faced due to the COVID-19 pandemic, which aligns with findings by Changwon Son in 2020 [13]. Key factors contributing to this burden included fears for health, uncertainty, and reduced social interaction, all of which intensified stress and burnout. These issues reflect trends observed in the existing studies and further hindered engagement in the research [11,20,21]. Survey findings from the National Institute of Health (NIH) Extramural Survey, including 45,348 researchers, indicated that mental and physical health is the number one factor negatively impacting the productivity of early career investigators, Hispanics, and African American respondents [20]. In addition, a cross-sectional study conducted by the Puerto Rico Public Health Association suggests that the well-being of a cohort from the Puerto Rican community has declined due to a lack of mental and social engagement [22].

The researchers implemented mental health support tools, such as virtual support groups and mental health virtual seminars, to address these challenges. This integration alleviated immediate emotional burdens and highlighted the importance of prioritizing research staff well-being in research design. This approach is consistent with recommendations from previous studies that emphasize the need for mental health considerations during crises [21].

Understanding the community’s needs was a pivotal component in navigating these challenges. Effective planning, recruitment strategies, and the distribution of incentives were critical to ensuring engagement in the study. Researchers could tailor their approaches to meet community expectations and needs by actively seeking feedback and adapting to evolving circumstances. This adaptability was essential for maintaining the research subjects’ involvement throughout the study. Moreover, adjustments to study protocols were imperative to maximize study subjects’ retention. Distributing incentives through mail and introducing a drive-through option gave the study subjects flexible choices that accommodated their comfort levels and preferences during the pandemic. These innovations enhanced the study subjects’ experience, met their community needs, and reinforced their commitment to the study.

Attendees raised concerns regarding accessibility and explored the potential applications of technology to mitigate the limitations faced. Digital platforms have played a crucial role in overcoming barriers related to study subjects’ recruitment and communication among research staff. Tools such as telephone calls, chat applications, social media, video conferencing, and all-in-one productivity applications have enabled connections with potential study subjects and research staff, despite the limitations imposed by the pandemic. These tools align with the major work technologies in COVID-19, as summarized by a review article from the State University of New York, Ringgold Standard Institution [11]. These digital tools not only facilitated communication but also streamlined the informed consent process. Furthermore, the flexibility of remote work has allowed researchers to adapt to study subjects’ varying availability. The ability to engage digitally has become essential in the current research landscape, as highlighted by the effective strategies employed by institutions [16,18] and the numerous benefits seen through the shift to online tools [19].

According to data from the NIH RePORTER at the University of Puerto Rico, research productivity increased from 2020 to 2021. Specifically, there was a 41.0% rise in publications, a 19.0% increase in grants, and a 6.0% growth in research projects [8]. The increase in scientific writing, new supplement applications, new funding opportunities, and new protocols developed during the pandemic reflect the resilience of the research community and its ability to adapt to challenging circumstances. The pandemic not only highlighted existing need for research but also spurred increased collaboration and innovation.

Finally, the feedback received regarding oral presentations and group discussions was overwhelmingly positive (Table 3 and Table 4). A significant majority of attendees agreed that these activities significantly contributed to their professional development. Additionally, many expressed a desire to engage in similar activities in the future. This response underscores the value of continuous learning and collaborative engagement in the research process and emphasizes fostering a supportive community among researchers. The positive reception affirms the potential for future initiatives that prioritize professional development within research settings.

The limitations of this study may significantly impact the findings and their generalizability. One key limitation is the small sample size, which could lead to a lack of statistical power and make it difficult to draw definitive conclusions. Additionally, the study’s focus on researchers and staff directly associated with the RCMI program introduces potential selection bias. Because the sample does not include personnel from the entire Campus, the perspectives and experiences of a broader range of individuals are omitted. This may result in findings that do not fully represent the diverse views and contexts of the entire institution, limiting the study’s applicability to a wider audience. Moreover, similar studies are essential in our population from a non-academic research perspective, taking into account the institutional and workplace challenges faced by research staff during the pandemic. Addressing these limitations in future research will be essential for obtaining more comprehensive and robust insights.

The findings from the retreat provide valuable insights for shaping future research strategies in similar institutions. They advocate for a community-centric approach to engage local populations, emphasize the adaptability of research protocols in response to unforeseen challenges, and promote the use of digital tools for enhanced collaboration and data collection. Additionally, the findings highlight the importance of data-driven decision-making to foster continuous learning and encourage cross-disciplinary collaboration to address complex issues. They also underscore the necessity of emergency preparedness to ensure research continuity in the face of disruptions, such as the COVID-19 pandemic. In summary, the retreat’s findings stress the importance of addressing community needs, adapting research protocols, and leveraging digital tools to navigate the challenges posed by unprecedented circumstances like the COVID-19 pandemic. By applying these insights, institutions can enhance their research capabilities, making them more resilient, relevant, and responsive to the needs of the communities they serve.

## 5. Conclusions

Research challenges during a healthcare emergency can significantly influence health disparities research. The COVID-19 pandemic and the associated restrictions aimed at curbing contagion adversely affected academic and research activities at the University of Puerto Rico, Medical Sciences Campus. Researchers, especially those involved with human subjects, adapted by amending IRB protocols and redirecting their focus towards alternative tasks, such as drafting academic papers and proposals. The recommendations highlight the necessity for flexible research protocols that can adjust to unforeseen circumstances and advocate for streamlined IRB processes. Furthermore, there is a call for enhanced digital collaboration tools to improve communication among researchers, as well as community-centric approaches that actively involve local populations in the research design. Institutions should also prioritize the development of comprehensive emergency preparedness plans, foster cross-disciplinary collaboration through regular workshops, and implement continuous learning frameworks to cultivate an adaptive and innovative research environment. The lessons learned on methodological and practical responses will assist in successfully addressing these challenges in the future.

## Figures and Tables

**Table 1 ijerph-22-00623-t001:** Quantitative results of the Unforeseen Challenges, Actions Taken, and Emerged Opportunities reported by the attendees via a survey assessment conducted at the RCMI-CCRHD scientific retreat (*n* = 20).

	Frequency	Percentage (%)
Unforeseen challenges		
Remote work	13	65.0
Diminished social interactions affecting mental health	9	45.0
Recruitment limitations	9	45.0
Actions taken		
Usage of digital platforms and technologies to connect with research staff and study subjects	14	70.0
Modifying protocols to sustain study participation	12	60.0
Knowing community needs	6	30.0
Emerged opportunities		
Become knowledgeable of technology tools and social media platforms	13	65.0
More time to work in protocols and publications	9	45.0
New funding and supplement opportunities	8	40.0
Identify mental health tools to support personnel	3	15.0

**Table 2 ijerph-22-00623-t002:** Qualitative results of the Actions Taken and Emerged Opportunities reported by the attendees via a survey assessment conducted at the RCMI-CCRHD scientific retreat (*n* = 5).

	Qualitative Results
Actions taken	Response 1. “Given the shift to remote work and a smaller team in the lab, we put together a schedule for everyone to come in a few days a week. This way, we can keep things organized by grouping tasks together. It’s a straightforward way to stay productive while adapting to the new situation. Plus, it helps us make the most out of our time in the lab and keeps things running smoothly”. Response 2. “During the height of the COVID-19 pandemic, we switched to online seminars and created work groups, chats, and meetings to connect with each other and share what we were going through. These virtual hangouts really helped us cope with all the stress. We talked about ways to handle tough situations, and many of us focused on staying active and taking care of our mental health. It was a tough time, but it showed how adaptable we can be when faced with challenges”. Response 3. “Strengthening better communication with PIs was key during the Pandemic. It fostered a sense of community among researchers and ensured that everyone remained aligned on goals and priorities. This experience highlighted the importance of maintaining open lines of communication”.
Emerged opportunities	Response 4. “It’s pretty clear now that diving into the online world is super important; we can move the world online. Virtual meetings have opened up new ways to work with researchers beyond the MSC. Getting the hang of different apps and digital tools is key to reaching a wider audience and improving education. We need to step up our digital communication skills to better engage with others and share what we know”. Response 5. “As the pandemic passes, it’s clear that refining our planning for research materials is essential. We must be proactive, ensuring we have alternative resources ready to avoid delays from IRB approvals when protocols change. By anticipating these challenges, we can dedicate more time to crucial tasks like data analysis and writing. Effective time management has become vital for advancing our work despite the obstacles”.

**Table 3 ijerph-22-00623-t003:** The oral presentations rating by attendees (*n* = 20).

How Do You Rate the Following Presentations?	Excellent (%)	Good (%)	Fair (%)
COVID-19 Vaccine Uptake Study (PR-COVACUPS)	80.0	20.0	0.0
The Impact of COVID-19 on Maternal Health in Puerto Rico	80.0	20.0	0.0
Communicating Science to Lay Audiences	85.0	15.0	0.0
Group Activity	65.0	30.0	5.0

**Table 4 ijerph-22-00623-t004:** Evaluation of attendees’ satisfaction related to the activity based on their degree of agreement with the statements (*n* = 20).

Indicate How Much You Agree or Disagree with the Following Statements Related to the Activity:	Completely Agree (%)	Agree (%)
The coordination of this activity was handled effectively	100.0	0.0
The allocated time for this activity was adequate.	80.0	20.0
The agenda for the activity was appropriate	80.0	20.0
The place where the activity was held was appropriate.	80.0	20.0
The activity contributed to my professional development.	75.0	25.0
I would participate again in activities like these	85.0	15.0

## Data Availability

The data is contained within the article and Appendix A.

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
