# Peer review of "Navigating Research Challenges: Collaborative Insights from a Research Retreat During a Healthcare Emergency in Puerto Rico"

_ijerph, 2025, doi:10.3390/ijerph22040623_

Round 1
Reviewer 1 Report (Previous Reviewer 2)
Comments and Suggestions for Authors The author did make corresponding changes to my last review opinion. But another line155:45.0%?Author Response
Please see the attachments.

Reviewer 2 Report (Previous Reviewer 3)
Comments and Suggestions for Authors
Dear Authors,
I have received a re-submission for this article. The authors made very few changes and opted to submit this paper as a brief report. A brief report should be a preliminary work and not a full paper. Again, I strongly suggest submitting this paper as a Letter to the Editor and not a full article. A full-length explanation will be based on my previous comments which I have re-attached below:
"Initially, I thought this article would be a review article regarding the tough changes during COVID-19 in terms of research. However, the introduction also touches on education, which is out of topic. Furthermore, it turns out to be a report for a retreat activity, which I think is not worthy of an article. Overall, I suggest this article be repackaged as a letter to the editor instead of an article, or instead write a review article on what happens in terms of research challenges during COVID-19 in Puerto Rico".
Author Response
Please see the attachment.

Reviewer 3 Report (New Reviewer)
Comments and Suggestions for Authors
Thank you very much for submitting your manuscript Navigating Research Challenges: Collaborative Insights from a Research Retreat During the COVID-19 Pandemic in Puerto Rico. Congratulations!
I will now proceed with a thorough review. What aspects should be improved for its revision and possible acceptance.
The abstract presents a clear structure and well-defined context, highlighting the challenges, strategies, and lessons learned. However, several key areas for improvement have been identified. Firstly, it is essential to explicitly state the research objective to provide clear direction and emphasize its relevance. Additionally, the methodological design should be described in greater detail, specifying whether a qualitative, quantitative, or mixed approach was used, along with the data analysis process. Furthermore, the conclusion should be strengthened to better reflect the study’s implications and the applicability of its findings. To enhance the clarity and rigor of the abstract, it is recommended to include a statement describing the employed methodology, explicitly define the research objective, and reinforce the conclusion with concrete recommendations based on the results obtained.
The introduction provides a well-contextualized overview of the impact of COVID-19 on Puerto Rico and academic research, supported by an extensive and well-referenced literature review. However, some areas require improvement. The introduction mentions challenges but lacks an explicitly stated research question. Additionally, the study's originality in comparison to previous research on the pandemic’s impact on academic research should be clarified. To enhance clarity and rigor, it is recommended to explicitly state the research question or hypothesis and clearly indicate how this study contributes new knowledge compared to existing literature.
Upon review, I would appreciate it if you could address the following question:
- What novel insights does this study provide in comparison to previous research on the impact of COVID-19 on academic research?
The Materials and Methods section adequately describes the retreat activity and data collection methods, as well as the exemption process for informed consent by the ethics committee. However, it presents several important areas for improvement. It is unclear whether the study follows a qualitative or mixed-methods design and how the researchers attending the retreat were selected (e.g., purposive sampling, convenience sampling). Additionally, there is a lack of information regarding the qualitative analysis, specifically the approach used, which should be improved, as well as how the questionnaires were developed and validated, and the type of coding applied. It is also necessary to clarify whether any form of data triangulation was conducted to enhance the validity of the analysis. To strengthen this section, it is recommended to explicitly state the study design, define the participant selection criteria, provide a detailed description of the analytical procedure used, and indicate whether any validation mechanisms for the qualitative analysis were applied, such as peer review or data triangulation, among others.
The results are well-structured and present percentages and frequencies that facilitate interpretation. However, there is no analysis of significant differences between groups of attendees, nor are textual quotes included, which are essential in qualitative studies. To improve this section, it is recommended to include examples of participants' responses to enhance the credibility of the findings and analyze potential differences between subgroups, such as principal investigators and research assistants. Additionally, refining and detailing the methodology—specifically the study design and qualitative analysis—will strengthen the results, providing greater scientific rigor and a more solid foundation for the findings.
The discussion appropriately contextualizes the findings in relation to previous studies and highlights the importance of resilience and adaptation in research. However, it remains largely descriptive, without critically addressing the results or exploring methodological limitations in depth. Additionally, it does not analyze how the findings may be applicable to other contexts. To improve this section, it is recommended to compare the results more critically with similar studies, reflect on potential limitations such as sample size and selection bias, and discuss how these findings can contribute to the development of future research strategies in similar settings.
The conclusion emphasizes the strategies learned and the importance of adaptation but is overly general and lacks concrete practical recommendations. Moreover, it does not provide clear suggestions for future research. To strengthen this section, it is recommended to reinforce recommendations for the scientific community and propose future research directions based on the findings, ensuring greater applicability of the study in practice.
At a general level, it is also recommended to review the bibliography format to ensure compliance with editorial requirements.
Round 2
Reviewer 2 Report (Previous Reviewer 3)
Comments and Suggestions for Authors
Dear Authors,
I appreciate that you took the time to look through my comments. Unfortunately, if you stand by your decision, so can I.
Author Response
Please see the attachment.

Reviewer 3 Report (New Reviewer)
Comments and Suggestions for Authors
I have reviewed the manuscript titled "Navigating Research Challenges: Collaborative Insights from a Research Retreat During the COVID-19 Pandemic in Puerto Rico" once again. Below is a detailed analysis based on the previous review comments and the adjustments made.
For submissions, it is necessary to follow the journal’s recommendations and highlight changes in a different color to facilitate an easy and quick differentiation of the manuscript.
Abstract It was recommended to explicitly state the study’s objective, provide a more detailed description of the methodological design, and strengthen the conclusion to better reflect the applicability of the findings. I see that the structure remains clear and well-organized. Although the study objective is mentioned, my recommendation is to explicitly formulate it. The methodology could be better detailed in terms of qualitative coding and data analysis, and the conclusion could be reinforced with more specific recommendations derived from the results.
Introduction It was requested to include an explicit research question and clarify the originality of the study compared to previous research. I maintain the recommendation to include a clear research question within the text and explicitly expand on what differentiates this study from previous research.
Materials and Methods It was unclear whether the study followed a qualitative, quantitative, or mixed design. Additionally, the participant selection process was not detailed, and information about qualitative analysis and data validation was missing. The recommended improvement is to explicitly state whether the sample was convenience or purposive and justify this choice. Include information on the validation of the questionnaires and coding, and mention whether data triangulation was conducted.
Results It was recommended to analyze significant differences between subgroups and to include direct quotes to support the qualitative findings. The recommended improvement is to analyze differences in responses according to the participants' roles and integrate direct quotes into the narrative of the results.
Discussion Previous comments: A more critical analysis was recommended in comparison with previous studies. Additionally, it was suggested to address methodological limitations and the applicability of the findings. The recommendation is to improve the critical reflection on methodological limitations and explore the applicability of the findings in other contexts.
Conclusion It was recommended to strengthen the recommendations and suggestions for future research. The recommendations remain general and do not translate into concrete actions. Therefore, it is recommended to propose specific recommendations for the scientific community and include suggestions for future studies.
General Conclusion The manuscript has improved, but it still requires adjustments in methodology, results analysis, and applicability. It is recommended to enhance the explicitness of the research question, provide a more detailed qualitative analysis, and offer more specific conclusions.
Author Response
Please see the attachment with response letter and highlighted changes

This manuscript is a resubmission of an earlier submission. The following is a list of the peer review reports and author responses from that submission.
Round 1
Reviewer 1 Report
Comments and Suggestions for Authors
After reviewing the submitted document, I would like to make several observations:
Firstly, the topic seems novel in relation to the challenges faced by research staff during the COVID-19 pandemic. It also has the potential to provide relevant insights for addressing future disasters of a similar nature. However, I have identified some areas for improvement in this manuscript.
Regarding the introduction, it is somewhat lengthy compared to the results section and dedicates a significant portion to discussing the impact of COVID-19 on education. This is barely addressed in the methodology and results, considering that the focus is on researchers.
Furthermore, the sample size is rather small for drawing more meaningful conclusions in the field of science. Additionally, although there were 20 participants, not all of them completed the required surveys, further reducing the level of participation.
In terms of the results, it would have been interesting to explore how the work mechanisms operated prior to the pandemic, how the unit had to adapt, and what the consequences were. Moreover, it would be helpful to explain the limitations encountered—for example, if someone was unable to access the necessary technology and how such issues were resolved.
Finally, in the discussion section, it would have been beneficial to explore findings from other studies on the challenges faced by other universities. Were there any universities that did not encounter problems? What solutions were proposed? A comparative discussion of the situation in relation to other contexts would have added depth to this section.
Reviewer 2 Report
Comments and Suggestions for Authors
1. Please modify the tables into a standard three-line table form.
2. Some spaces in the text are improper, please check and modify them carefully.
3. Authors can use the 1-digit decimal to make it consistent with the percentage.
4. Abstract section, some mythologies should be descried briefly in abstract section, contribute the readers to understand the design of the study.
5. The discussion section need to further improve, and make sure the context is consistent with purpose of objective and main findings.
6. A Lot of work needs to be done before publication.
Comments on the Quality of English Language
1. The author should pay more attention to language, especially spelling, some mistakes in the manuscript, such as line 117 “lessos”, line 133 “acepted”, line 134 “radomly”, and so on. Please check the whole manuscript carefully.
Reviewer 3 Report
Comments and Suggestions for Authors
Dear Authors,
I have read the manuscript on navigating research challenges in Puerto Rico.
Initially, I thought this article would be a review article regarding the tough changes during COVID-19 in terms of research. However, the introduction also touches on education, which is out of topic. Furthermore, it turns out to be a report for a retreat activity, which I think is not worthy of an article in itself.
What happens during oral presentation at that time surely does not reflect upon research challenges during COVID-19, which I am also confused as why it is being reported.
Overall, I suggest this article to be repackaged as a letter to editor instead of an article, or instead write a review article on what happens in terms of research challenges during COVID-19 in Puerto Rico
